# Prognostic Value of the miR-17~92 Cluster in Chronic Lymphocytic Leukemia

**DOI:** 10.3390/ijms24021705

**Published:** 2023-01-15

**Authors:** Sylwia Chocholska, Michał Zarobkiewicz, Agata Szymańska, Natalia Lehman, Justyna Woś, Agnieszka Bojarska-Junak

**Affiliations:** 1Department of Haematooncology and Bone Marrow Transplantation, Medical University of Lublin, 20-080 Lublin, Poland; 2Department of Clinical Immunology, Medical University of Lublin, 20-093 Lublin, Poland; 3Department of Clinical Transplantology, Medical University of Lublin, 20-093 Lublin, Poland

**Keywords:** CLL, microRNA, miR-17∼92-cluster, prognostic markers

## Abstract

The aim of this study was to investigate the expression of miR-17∼92 cluster members in chronic lymphocytic leukemia (CLL) patients. Six microRNAs (miRNAs)—miR-17, miR-18a, miR-19a, miR-19b-1, miR-20a, and miR-92a-1—very poorly characterized in CLL patients, were chosen for the study to consider their possible role as cancer biomarkers. It is currently unclear to which extent miR-17~92 expression is related to other routinely measured CLL markers, and whether the findings can be of any clinical significance. To achieve this goal, we report the expression levels of these miRNAs detected by RT-qPCR in purified CD19+ B lymphocytes of 107 CLL patients and correlate them with existing clinical data. The study provides new evidence regarding the heterogeneity of miR-17~92 cluster members’ expression in CLL patients. Higher miR-17-5p expression was associated with unfavorable prognostic factors (i.e., 17p and 11q deletions, CD38 and ZAP-70 expression). On the other hand, miR-19a, miR-20a, and miR-92a-1 negatively correlated with these adverse factors. The presence of del(13q) as a sole aberration was associated with a significantly lower miR-17-5p as well as higher miR-19a-3p and miR-92a-1-5p expression compared to patients carrying unfavorable genetic aberrations. Particularly, miR-20a could be considered an independent favorable prognostic factor. In a multivariate analysis, high miR-20a expression remained an independent marker predicting long TTT (time to treatment) for CLL patients.

## 1. Introduction

Chronic lymphocytic leukemia (CLL) is a malignancy characterized by clonal proliferation and accumulation of CD5+/CD19+/CD23+ B lymphocytes in peripheral blood, bone marrow, and secondary lymphoid tissues. CLL displays a heterogeneous spectrum of biological features and clinical manifestations. Due to clinical interindividual heterogeneity of CLL cases, identification of prognostic factors is of great importance in both predicting the course of the disease as well as selecting the optimal therapy [1,2,3,4].

The most clinically significant markers of poor disease outcome are the lack of mutations in the immunoglobulin heavy chain variable region locus (*IGHV*), deletion of chromosome 11q or 17p, as well as high expression of ZAP-70 (zeta-associated protein 70) and CD38 [1,2,3]. Moreover, recurrent genetic abnormalities, such as *TP53*, *ATM*, *NOTCH1*, *SF3B1*, *BIRC3*, and *MYD88* mutations are associated with disease prognosis, and some of them may influence treatment decisions [3,5,6]. Among different biomarkers, microRNAs (miRNAs) have appeared as new diagnostic and therapeutic biomarkers in CLL patients [7,8,9,10]. MiRNAs are an evolutionarily conserved group of small noncoding RNA molecules that regulate the expression of many genes. In the case of humans, the miRNA is most often 22 nucleotides long, though molecules ranging in length from 19 to 25 nucleotides have been described. MiRNAs are engaged in the regulation of a variety of biological processes, such as cell differentiation, proliferation, apoptosis, tumorigenesis, and angiogenesis [11]. MiRNAs can function either as oncogenes or tumor suppressors in various cancers such as CLL. Deregulation of miRNAs occurs in many types of human cancers, such as breast, ovarian, colon, and hematological cancers (AML, ALL, CML, CLL). Some of them, such as miR-15a, miR-16-1, miR-155, miR-181b, and members of the miR-17~92 cluster, play key roles in CLL onset and progression [7,9,12].

The miR-17~92 cluster (also termed oncomiR-I) comprises six mature miRNA members: miR-17, miR-18a, miR-19a, miR-19b-1, miR-20a, and miR-92a-1 [11]. It has been demonstrated that BCR signals can modulate the expression of microRNAs. Evidence that BCR response in aggressive unmutated *IGHV* genes CLL is accomplished through upregulation of miR-17∼92 has been found. The BCR-driven induction of miR-17∼92 in unmutated *IGVH* cases has been described as a possible regulator of B-cell proliferation/survival by downregulating antiproliferative and/or proapoptotic genes [13]. The aforementioned data prompted us to assess a prognostic potential of a miRNA-17~92 cluster in CLL patients.

The aim of this study was to investigate the expression of miR-17∼92 cluster members in CLL patients. Six miRNAs—miR-17, miR-18a, miR-19a, miR-19b-1, miR-20a, and miR-92a-1—very poorly characterized in CLL patients, were chosen for the study to consider their possible role as cancer biomarkers. It is currently unclear to which extent miR-17~92 expression is related to other routinely measured CLL markers, and whether the findings can be of any clinical significance. To achieve this goal, we report the expression levels of these miRNAs detected by RT-qPCR in purified CD19+ B lymphocytes of CLL patients and correlate them with existing clinical data.

## 2. Results

### 2.1. Patients’ Characteristics

The study included 107 treatment-naive patients (aged 37–85 years; median, 67 years) diagnosed with CLL according to International Workshop on Chronic Lymphocytic Leukemia (IWCLL) criteria [14,15]. All of CLL cases (100%) included in the analysis were CD5-positive and contained CD19+/CD5+ population (median (IQR; interquartile range) 79.05% (70.47–86.76%)). The study group consisted of 61 (57%) women and 46 (43%) men. The clinical stage was determined according to the Rai classification system [16]. Patients with Rai stage 0 were classified into a low-risk group (46.7%), patients of stages I/II were considered to have an intermediate risk (40.2%), and patients of stages III and IV were classified as a high-risk group (13.1%). For all samples, clinical and biological information, such as Rai stage, CD38 and ZAP-70 expression, need for treatment, and cytogenetic abnormalities were available. In 81 patients (75.7%), there was a complete concordance of ZAP-70 and CD38 expression, i.e., 23 patients (21.5%) were positive for ZAP-70 and CD38, and 58 patients (54.2%) showed the ZAP-70−/CD38− phenotype. Additionally, 7 patients (6.5%) were characterized by the ZAP-70+/CD38− phenotype and 19 patients (17.8%) showed the ZAP-70−/CD38+ phenotype. Moreover, the percentage of CD19+/CD5+/CD38+ cells positively correlated with the percentage of leukemic cells expressing ZAP-70 (r = 0.486; *p* < 0.0001). I-FISH analysis was performed in all cases using the probes for the 13q14 (DLEU1), 11q22 (ATM), and 17p13 (TP53) regions and trisomy 12 (CEP 12). Characteristics of CLL patients at the time of diagnosis is summarized in Table 1.

### 2.2. Expression of the miR-17∼92 Cluster in Different Risk Groups

We found a significantly higher (*p* < 0.05) miR-17-5p expression level in the CLL group with a high risk of progression (stage III/IV) as compared to the low-risk group, i.e., stage 0 (Table 2). On the other hand, significantly (*p* < 0.05) higher expression levels of miR-19a-3p and miR-20a-5p were observed in the low-risk group as compared to the high-risk group (Table 2). The relative expression levels of miR-18a-5p, miR-19b-1-5p, and miR-92a-1-5p were not significantly different between risk groups.

There was a weak positive correlation between WBC count and the miR-17-5p expression in leukemic B cells (r = 0.253; *p* < 0.05) and miR-18a-5p expression (r = 0.242; *p* < 0.05). The analysis also revealed a significant weak positive correlation between lymphocyte count and miR-17-5p (r = 0.208; *p* < 0.05) and miR-18a-5p expression (r = 0.271; *p* < 0.05). Moreover, we have found a negative correlation between miR-92a-1-5p expression in B cells and WBC (r = −0.226; *p* < 0.05) or lymphocyte count (r = −0.212; *p* < 0.05). However, none of the other miR-17∼92 members significantly correlated with the WBC or lymphocyte count. Regarding the serum LDH and β2-microglobulin levels or patients’ age, no significant correlations were found.

### 2.3. miR-17∼92-Cluster Expression and Genetic Aberrations

The group of patients carrying 11q22.3 deletion, trisomy 12 and/or 17p13.1 deletion showed significantly higher miR-17-5p expression in B lymphocytes compared to patients without these adverse aberrations (median (IQR), 1.299 (0.595–2.009) vs. 0.680 (0.141–1.360), *p* < 0.01; Figure 1). Additionally, the expression level of miR-17-5p in the group of patients with these unfavorable genetic aberrations was higher (*p* < 0.05) compared to the group with del(13q14) (median (IQR), 0.479 (0.198–1.343)) (Figure 1). On the other hand, miR-19a-3p and miR-92a-1-5p relative expressions in B cells were significantly higher in patients with del(13q14) compared to patients carrying 11q22.3 deletion, trisomy 12 and/or 17p13.1 deletion or to patients without these adverse aberrations (*p* < 0.05, Figure 1). Such differences were not found for miR-18a-5p, miR-19b-1-5p, and miR-20a-5p (*p* > 0.05, Figure 1).

### 2.4. miR-17∼92-Cluster Expression and Chromosome 13q14 Deletion

The deletion of the 13q14 region is the most common abnormality found in CLL cases. Isolated 13q deletion is related with a better prognosis than combined aberrations involving other chromosomes [17]. In our study, 45 out of 107 CLL patients (42%) had del(13q14) without other cytogenetics abnormalities, i.e., del(11q22.3), del(17p13.1), or +12. As described earlier, the presence of del(13q14) in FISH analysis was associated with a significantly lower miR-17-5p and higher miR-19a-3p and miR-92a-1-5p expression compared to patients carrying unfavorable genetic aberrations (Figure 1). In addition, patients with monoallelic 13q deletion had lower expression of miR-17-5p in leukemic lymphocytes compared to biallelic deletions (*p* < 0.05, Figure 2). On the other hand, patients with biallelic del(13q14) showed higher miR-20a-5p expression in leukemic B cells (*p* < 0.05, Figure 2). Such differences were not observed for miR-18a-5p, miR-19a-1-5p, miR-19b-1-5p or miR-92a-1-5p (*p* > 0.05, Figure 2).

### 2.5. miR-17∼92-Cluster Expression and ZAP-70 Expression

We found a significantly higher miR-17-5p expression level in ZAP-70-positive CLL patients as compared to ZAP-70-negative ones (*p* < 0.05). On the other hand, significantly higher expression levels of three cluster members, i.e., miR-19a-3p (*p* < 0.05), miR-20a-5p (*p* < 0.001), and miR-92a-1-5p (*p* < 0.01) were observed in a ZAP-70-negative group as compared to ZAP-70-positive patients (Figure 3). The relative expression levels of miR-18a-5p and miR-19b-1-5p were not significantly different between the ZAP-70-positive and ZAP-70-negative groups (Figure 3).

### 2.6. miR-17∼92-Cluster Expression and CD38 Expression

We found a significantly higher (*p* < 0.05) miR-17-5p expression level in CD38-positive patients as compared to CD38-negative ones (*p* < 0.05). On the other hand, significantly higher expression levels of miR-19a-3p (*p* < 0.01), miR-20a-5p (*p* < 0.05), and miR-92a-1-5p (*p* < 0.05) were observed in a CD38-negative group (Figure 4). The relative expression levels of miR-18a-5p and miR-19b-1-5p were not significantly different between CD38-positive and CD38-negative groups (Figure 4).

### 2.7. miR-17∼92-Cluster Expression and Clinical Outcome of CLL Patients

The median follow-up time was 45 months (range 5–72 months). During the follow-up period, 45 patients (42.1%) required treatment. The miR-17-5p expression measured at the time of diagnosis was significantly higher in patients requiring therapy during the observation period (*p* < 0.05) (Figure 5).

Regarding treatment response, complete remission (CR) was achieved in 10 CLL patients (22.2%), partial remission (PR) in 9 patients (20%), stable disease (SD) in 10 patients (22.2%), and disease progression (PD) was observed in 16 patients (35.6%). In patients with PD, the expression of miR-17-5p in CLL cells was significantly higher than in those with CR or PR (*p* < 0.05). Contrarily, miR-20a-5p and miR-92a-1-5p expression was significantly lower in the patients with PD than in those with CR or PR (*p* < 0.05) (Table 3). Seven CLL-related deaths occurred during the observation period. Patients who died showed higher miR-17-5p expression than patients with CR/PR (*p* < 0.05). Moreover, higher miR-20a-5p (*p* < 0.05) and miR-92a-1-5p (*p* < 0.05) expressions were found in CLL patients with CR and/or PR than in patients who died.

Based on the ROC analysis, we determined the optimal thresholds for miRNAs expression that were associated with ZAP-70 expression:-The cutoff value for miR-17-5p expression was found to be: >0.748 (AUC, 0.752; 95% confidence interval [CI], 0.656–0.842; *p* < 0.0001; Figure 6a).-The cutoff for miR-19a-3p expression was found to be: <1.151 (AUC, 0.653; 95% CI, 0.546–0.760; *p* < 0.01; Figure 6b).-The cutoff for miR-20a-5p expression was found to be: <1.629 (AUC, 0.719; 95% CI, 0.612–0.825; *p* < 0.0001; Figure 6c).-The cutoff for miR-92a-1-5p expression was: <0.883 (AUC, 0.666; 95% CI, 0.547–0.725; *p* < 0.01; Figure 6d).-For miR-18a-5p and miR-19b-1-5p, it was not possible to determine the optimal cutoff point with a significance of *p* < 0.05.

Next, the group of 107 CLL patients was divided according to the determined cut-off values of miR-17-5p, miR-19a-3p, miR-20a-5p, and miR-92a-1-5p expression levels into low and high groups.

We observed that low miR-20a expression (<1.629) was strongly associated with shorter TTT (time to treatment) (*p =* 0.002) (Table 4, Figure 7). Similarly, miR-19a-3p expression in B cells significantly (*p =* 0.006) impacted the TTT (Table 4, Figure 7). Low expression of miR-19a-3p (<1.151) was identified in CLL patients with shorter TTT with a mean of 41 months vs 49 months in patients with high miR-19a-3p expression (Table 4, Figure 7). We also analyzed the relationship between miR-17-5p and miR-92a expression and TTT. However, the difference did not reach statistical significance (Table 4, Figure 7). In a multivariate analysis, when considering age, β2M, ZAP-70, CD38 and cytogenetic aberrations, low miR-20a expression remained an independent marker predicting short TTT for CLL patients (HR, 2.96; 95% CI, 1.38–6.38; *p* = 0.006) (Table 4). However, in univariate and multivariate analysis, OS (overall survival) seemed not to be influenced by miR-17-5p, miR-19a-3p, miR-20a-5p, and miR-92a-1-5p expression levels in leukemic B cells (*p* > 0.05).

## 3. Discussion

The discovery of the miR-17∼92 cluster took place in 2004 [18], and only a year later its oncogenic character, based on the B-cell lymphoma model, was determined [19]. Therefore, it is also referred to as the ‘oncomiR1’ [20]. The miR-17∼92 cluster maps to human chromosome 13 and encodes six individual miRNAs (miR-17, miR-18a, miR-19a, miR-20a, miR-19b, and miR-92a) with a host gene *MIR17HG* [20]. The miR-17∼92 cluster can be highly expressed in a wide range of tumor cells and types of cancer such as lung, breast, pancreatic, prostate and thyroid cancer, as well as lymphomas [11,20]. Most of the studies have analyzed the carcinogenic potential of the miR-17∼92 cluster, but it has to be remembered that the cluster can also have an antitumor function [11]. Some of its members also have a significant diagnostic value for the detection of secondary central nervous system involvement in B-cell non-Hodgkin lymphoma vs primary central nervous system lymphomas [21]. Though it has great potential in the prediction of various cancer types’ prognosis [22], an exact prognostic value of the highly expressed miR-17∼92 cluster in CLL remains controversial [23].

The miR-17∼92 cluster should be considered as a complex of miRNA individuals with distinctive properties. It remains unclear how different members of the cluster contribute to the oncogenic activity. Thus, in our study, all currently known cluster components were analyzed in relation to commonly used clinical markers (e.g., ZAP-70, sole del(13q14)) and to therapy requirements. The oncogenic miR-17∼92 cluster is located on chromosome 13q. An interstitial deletion of this region is the most common cytogenetic change in CLL and isolated 13q deletion is related with a better prognosis than combined aberrations involving other chromosomes. Additionally, as a sole abnormality, 13q deletion is associated with an indolent disease [24]. However, it is observed that the clinical course of CLL cases with del(13q) is quite heterogeneous and the reason for this clinical diversity has not yet been established [17]. In our study, 42% of CLL patients had del(13q14) without other cytogenetic abnormalities, i.e., del(11q22.3), del(17p13.1), or +12. The presence of del(13q14) as a sole aberration was related with a significantly lower miR-17-5p and higher miR-19a-3p, and miR-92a-1-5p expression compared to patients carrying unfavorable genetic aberrations. Durak Aras et al. (found that biallelic 13q deletions determine a bad prognosis. The authors mentioned that biallelic del(13q) is aggressive [17]. However, others did not find significant differences in the baseline characteristics and clinical outcomes among CLL patients with monoallelic or biallelic deletions [25]. When we investigated whether there was a difference between the monoallelic or biallelic groups in the miR-17∼92-cluster expression, we found that patients with monoallelic 13q deletion had a lower expression of miR-17-5p in leukemic lymphocytes compared to biallelic deletions. On the other hand, patients with biallelic del(13q14) showed higher miR-20a-5p expression in leukemic B cells. Moreover, it is interesting that in univariate and multivariate analysis, low miR-20a expression remained an independent marker predicting short TTT for CLL patients.

It should be noted that the miR-17∼92 cluster is involved in the development and maturation of B cells [26]. Moreover, miR-17∼92 expression is strongly induced in CLL cells by CD40L stimulation [27]. The miR-17∼92 cluster is a downstream regulator of the *MYC* network [28]. What is unique for the miR-17∼92 cluster, *MYC* expression leads to its activation, whereas other miRNAs are repressed in B-cells [29,30]. The functional heterogeneity of the miR-17∼92 cluster is evidenced by many scientific reports. For example, two cluster members, i.e., miR-17 and miR-19b-1, play an important role in promoting B-cell proliferation, protecting B-cells from death, supporting IFNγ production and suppressing T-cell differentiation [20]. Furthermore, miR∼17-92, but especially miR-20a, is capable of significant silencing of MIC-A/B and ULBPs, important ligands for NKG2D [31]. This may significantly lower the cytotoxic response of γδ T and NK cells as both populations widely express NKG2D [32].

It is worth noting that the expression of the miR-17∼92 cluster in CLL patients in our study was diverse. MiR-18a-5p, miR-19b-1-5p, and miR-92a-1-5p expression levels were not different between risk groups. However, the miR-17-5p expression increased with the disease stage. On the other hand, higher expression levels of miR-19a-3p and miR-20a-5p were observed in the low-risk group. That is in agreement with the results of Selven et al. who showed a favorable prognosis for colorectal cancer patients with high miR-20a-5p expression [33]. Similarly, Farzadfard et al. reported higher circulating miR-19a-3p in CLL patients than in healthy subjects [2]. Moreover, Khalifa et al. showed significant overexpression of all miR-17∼92 cluster members in CLL patients in comparison to the control group [34]. Several studies suggest that miR-19a-3p may play an important role in the pathogenesis and progression of CLL [2,34,35]. Other studies have found that miR-20a-5p and miR-19a-3p are overexpressed in many cancers, although their function is still unknown [36]. However, there is no doubt that miR-20a-5p expression is crucial in cancer development and outcome, which may suggest its particular importance as a biomarker, and in the future it may contribute to the development of targeted drugs. Our study confirmed an aberrant cellular miRNA expression profile in leukemic B cells. Expression of three miR-17∼92 cluster members, i.e., miR-19a-3p, miR-20a-5p and miR-92a-1-5p correlate well with prognostic factors, including ZAP-70 and CD38 expression. The expression heterogeneity is worth underlining since the expression of miR-19a-3p, miR-20a-5p, and miR-92a-1-5p was higher in the ZAP-70 and CD38-negative group as compared with ZAP-70 and CD38-positive patients. On the other hand, the miR-17-5p expression level in the ZAP-70-positive CLL patients was higher as compared with ZAP-70-negative ones.

Moreover, higher expression of miR-19a-3p and miR-92a-1-5p was observed in patients with favorable cytogenetic changes, i.e., del(13q14) compared with CLL patients carrying 11q22.3 deletion, trisomy 12 and/or 17p13.1 deletion or a group without these adverse aberrations. In the current study, we have again found a correlation between an increase in the miR-17-5p expression and clinical markers of poor CLL outcome. Due to the miR-17∼92 cluster localization on chromosome 13, our findings regarding miR-17 expression and del(13q14) are especially interesting. The presence of the 13q deletion correlated with a significantly lower miR-17-5p expression. What is interesting, in the current study the lower expression of miR-17-5p was noted for monoallelic 13q deletion vs biallelic deletion. When the 13q chromosome deletion is a sole abnormality, monoallelic or biallelic 13q deletion seems to be associated with the favorable outcome [17]. Nevertheless, the differences in the prognosis between mono- vs. biallelic CLL patients still remain unsettled [37]. However, the current results of the miR-17 expression could be related to the worse outcome of the biallelic CLL patients [38].

The data obtained in the current study are nonconcordant with those reported by Khalifa et al. who demonstrated a significantly lower expression of miR-17-5p and CD38-negative and ZAP-70-negative group in CLL patients [34]. Our data is similar to Bomben et al.’s results who demonstrated significantly higher expression levels of miR-17-5p in ZAP-70-positive CLL patients [39]. In addition, miR-17-5p has been shown to be overexpressed in acute myeloid leukemia and could be considered as an independent negative prognostic factor [40]. On the other hand, Moussay et al. found that levels of circulating miR-20a-5p correlated with the disease stage, which was similar to the value obtained using ZAP-70 directly [41]. The authors suggest that the levels of miRNAs are strongly linked to cellular ZAP-70 expression status and the level of miR-20a-5p in plasma can be used as a marker for CLL patient management, replacing perhaps the ZAP-70 expression status analysis in leukemic B cells. CLL patients with either a lower level of miR-20a-5p in plasma or ZAP-70+ expression status in CLL cells will need more aggressive and earlier treatment [41].

In the present study, we showed higher expression levels of three cluster members, i.e., miR-19a-3p, miR-20a-5p and miR-92a-1-5p in CLL patients who did not require treatment during the observation period as compared with patients requiring therapy. We also demonstrated higher miR-20a-5p and miR-92a-1-5p expression levels in CLL patients with CR and/or PR than in patients who died. The results of the current study are consistent with those of Selven et al. that have shown that high expression levels of miR-20a-5p in tumor tissue are independent indicators of favorable disease-specific survival in colon cancer [33]. Likewise, Papageorgiou et al. reported that high miR-92a-3p expression predicts significantly prolonged OS (overall survival) of CLL patients [10]. On the other hand, Moussay et al. tested the possibility of using extracellular miRNAs in plasma to assess the disease severity and the time interval from diagnosis to treatment [41]. In their study, miR-20a-5p was found to correlate with diagnosis-to-treatment time in CLL and thus can potentially serve as a blood biomarker [41].

## 4. Materials and Methods

### 4.1. Patients and Samples

The study group comprised 107 patients diagnosed with CLL (46 males and 61 females, aged 37–85 years; median, 67 years), based on criteria from the International Workshop on Chronic Lymphocytic Leukemia (IWCLL) [14]. All subjects were newly diagnosed and treatment-naive. Peripheral blood (PB) samples were collected at the time of diagnosis, prior to any anticancer therapy. CLL patients were recruited between March 2016 and December 2021 in the Department of Hematooncology and Bone Marrow Transplantation of the Medical University of Lublin (Lublin, Poland). Clinical stage was determined according to the Rai classification system [16]; 50 patients were Stage 0, 18 patients were Stage I, 25 patients were Stage II, 5 patients were Stage III and 9 patients were Stage IV. Patients with Rai stage 0 were classified as a low-risk group, patients of stages I/II were considered to have an intermediate risk, and patients of stages III and IV were classified as a high-risk group. Characteristics of CLL patients at the time of diagnosis is summarized in Table 1.

The study was approved by the Ethics Committee of the Medical University of Lublin. Written informed consent was obtained from all patients with respect to the use of their blood for scientific purposes.

### 4.2. PBMCs Isolation

Peripheral blood mononuclear cells (PBMCs) were isolated by density gradient centrifugation (Biocoll; Merck Biochrom, Berlin, Germany) at 400× *g* for 20 min and washed twice in phosphate buffered saline (PBS without Ca^2+^ and Mg^2+^ ions; Merck Biochrom, Berlin, Germany) at 300× *g* for 5 min. PBMCs were then subjected to both cytogenetic as well as molecular tests.

### 4.3. I-FISH Analysis

PBMCs were cultured for 24 h in RPMI 1640 Medium (Merck Biochrom, Berlin, Germany) with 10% fetal bovine serum (FBS; Merck Biochrom, Berlin, Germany) and without mitogen stimulation. After hypotonic treatment with 0.075M potassium chloride solution and fixation with Carnoy’s solution (3:1 methanol-acetic acid mixture), cell suspensions were dropped onto microscopic slides and used directly for interphase fluorescence in situ hybridization (I-FISH) analysis. The following standard set of four commercially available FISH probes was used: Vysis LSI TP53 SpectrumOrange/ATM SpectrumGreen Probes and Vysis LSI D13S319 (DLEU1) SpectrumOrange/13q34 (LAMP1) SpectrumAqua/CEP 12 SpectrumGreen Probe (Vysis CLL FISH Probe Kit; Abbott GmbH, Wiesbaden, Germany). At least 200 nuclei were analyzed for each probe. The cutoff levels for positive results, determined for normal controls, were 2.5% (mean ± SD).

### 4.4. RNA Preparation and microRNA Expression Analysis

All molecular tests were performed on purified CD19+ B lymphocytes, sorted by positive magnetic separation using CD19 MicroBeads (Cat No.: 130-050-301, Miltenyi Biotec, Bergisch Gladbach, Germany) and the Magnetic Antibody Cell Separation (MACS) method in accordance with manufacturer’s recommendations. Total RNA was isolated using the mirVana miRNA Isolation Kit (Cat No.: AM1560, ThermoFisher Scientific, Waltham, MA, USA) according to the manufacturer’s protocol. The purity of isolated CD19+ B cells was assessed by flow cytometry and was usually greater than 95%. Moreover, over 92% of selected B cells showed the CD19+/CD5+ phenotype.

By combining acid phenol: chloroform extraction followed by RNA binding on silica filters, total RNA enriched for small RNAs, including micro RNA (miRNA), was obtained. RNA quality and quantity were assessed spectrometrically (BioSpec nano-spectrophotometer, Shimadzu Biotech, Kyoto, Japan) and 10 ng of total RNA was used in each reverse transcription (RT) reaction.

MicroRNA expression was analyzed using the TaqMan MicroRNA Assays (Applied Biosystems, Foster City, CA, USA) and the Real-Time Polymerase Chain Reaction (RT-PCR) method according to the manufacturer’s protocol. TaqMan MicroRNA RT Kit (Cat No.: 4366596, ThermoFisher Scientific, Waltham, MA, USA) was used for reverse transcription (RT). The reaction was carried out on a total RNA matrix with miRNA-specific stem–loop primers. The resulting complementary DNA (cDNA) was used for microRNA expression analysis, carried out by using TaqMan Universal PCR Master Mix II (Cat No.: 4440040, ThermoFisher Scientific, Waltham, MA, USA) and specific TaqMan probes. The following set of six assays was used: hsa-miR-17-5p (Assay ID 002308), hsa-miR-18a-5p (Assay ID 002422), hsa-miR-19a-3p (Assay ID 000395), hsa-miR-20a-5p (Assay ID 000580), hsa-miR-19b-1-5p (Assay ID 002425) and hsa-miR-92a-1-5p (Assay ID 002137). RT-qPCR reactions were performed using the Applied Biosystems 7300 Real-Time PCR System (Thermo Fisher Scientific, Applied Biosystems, Inc., Waltham, MA, USA). The level of the miR-17∼92a cluster expression was normalized to hsa-miR-16-5p (Assay ID 000391) as endogenous control and the relative expression of miRNAs was calculated using the 2^−∆∆Ct^ formula.

### 4.5. Flow Cytometric Analysis of CD38 and ZAP-70 Expression in CLL Cells

CD19+/CD5+ cells were stained for CD38 and ZAP-70 expression (as described previously [42,43]). A cutoff point for ZAP-70 positivity in leukemic cells was >20%. Patients with CD38 expression higher than 30% were classified as CD38 positive.

### 4.6. Statistical Analysis

The Kruskal–Wallis test with Dunn correction or U Mann–Whitney test were used for comparative analysis of the variables. The data are presented as the median and interquartile range (IQR). The Spearman rank correlation coefficient was used in correlation tests. Kaplan–Meier curves were used to compare time to treatment (TTT) and overall survival (OS) between categorical groups. The log-rank test was used to determine differences between groups. OS and TTT were determined from the date of diagnosis until the last follow-up/death and the date of initial treatment, respectively. Cox regression analysis was constructed to determine the hazard ratio (HR). All variables found to be significant at a level of *p* < 0.05 in the univariate analyses were included in multivariate analysis performed using Cox proportional hazards model. ROC (receiver operating characteristics) analysis and Youden index method were used to calculate the most significant cutoff value of miRNA expression that best distinguished ZAP-70-positive and ZAP-70-negative cases. AUC (area under the curve) was also estimated. Differences were considered statistically significant with a *p*-value < 0.05.

Statistical analysis was performed with Statistica 13 PL (StatSoft, Cracow, Poland). Graphs were prepared using GraphPad Prism 9 (GraphPad Software, San Diego, CA, USA) and Statistica 13 PL.

## 5. Conclusions

This study provides new evidence regarding the heterogeneity of miR-17~92 cluster members’ expression in CLL patients. MiR-17-5p is closely associated with adverse prognostic factors (i.e., 17p and 11q deletions, CD38 and ZAP-70 expression). On the other hand, miR-20a-5p, closely related with low ZAP-70 and CD38 expression, with lack of unfavorable cytogenetic aberrations and with the presence of del(13q) could be considered an independent favorable prognostic factor.

## Figures and Tables

**Figure 1 ijms-24-01705-f001:**
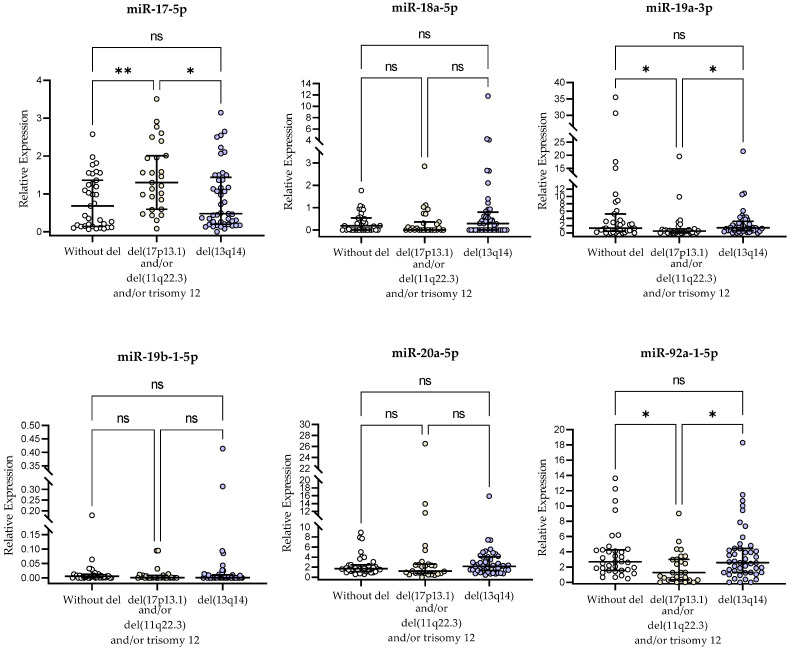
miR-17∼92-cluster expression in CLL patients carrying unfavorable genetic aberrations (del(11q22.3), del(17p13.1), trisomy 12), in patients with del(13q14) and patients without these genetic aberrations. Scatter plots present raw data. The central line shows the median. “Whiskers” represent from the first quartile to the third quartile (IQR, interquartile range). * *p* < 0.05, ** *p* < 0.01; ns, not significant; the *p*-value was calculated using the Kruskal–Wallis test with post-hoc Dunn’s correction.

**Figure 2 ijms-24-01705-f002:**
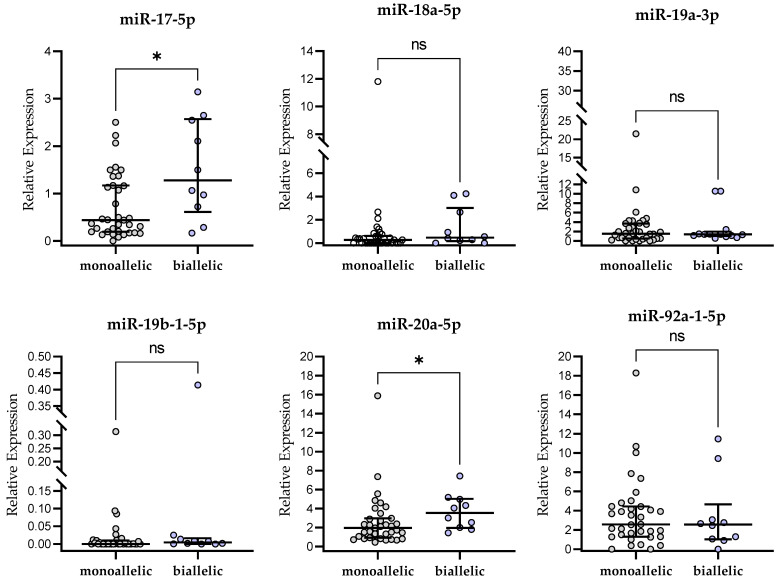
miR-17∼92-cluster expression in CLL patients with monoallelic and biallelic del(13q14). Scatter plots present raw data. The central line shows the median. “Whiskers” represent from the first quartile to the third quartile (IQR, interquartile range). * *p* < 0.05; ns, not significant; the *p*-value was calculated using the U Mann–Whitney test.

**Figure 3 ijms-24-01705-f003:**
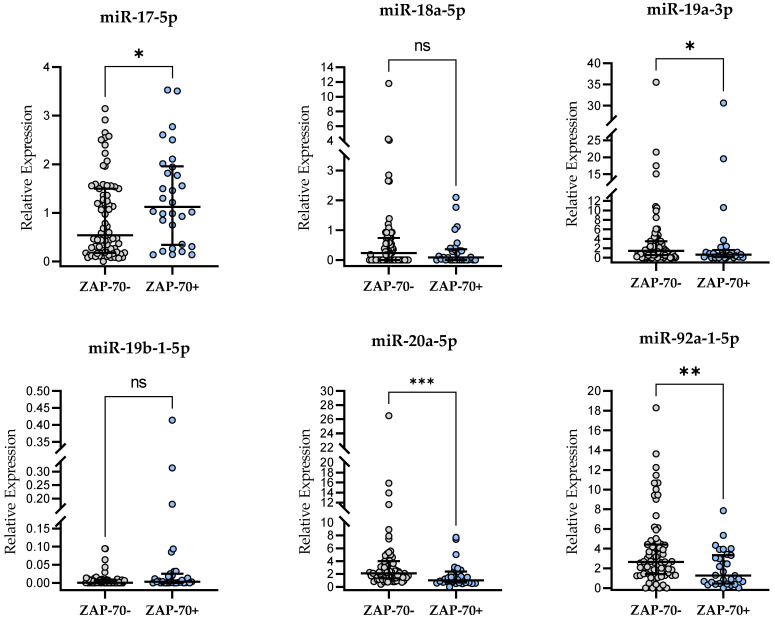
miR-17∼92-cluster expression in ZAP-70-positive and ZAP-70-negative patients. The central line shows the median and “whiskers” represent IQR. Scatter plots present raw data. The central line shows the median. “Whiskers” represent from the first quartile to the third quartile (IQR, interquartile range). * *p* < 0.05, ** *p* < 0.01; *** *p* < 0.001; ns, not significant; the *p*-value was calculated using the U Mann–Whitney test. A cutoff point for ZAP-70 positivity in CD19+/CD5+ cells was ≥20%.

**Figure 4 ijms-24-01705-f004:**
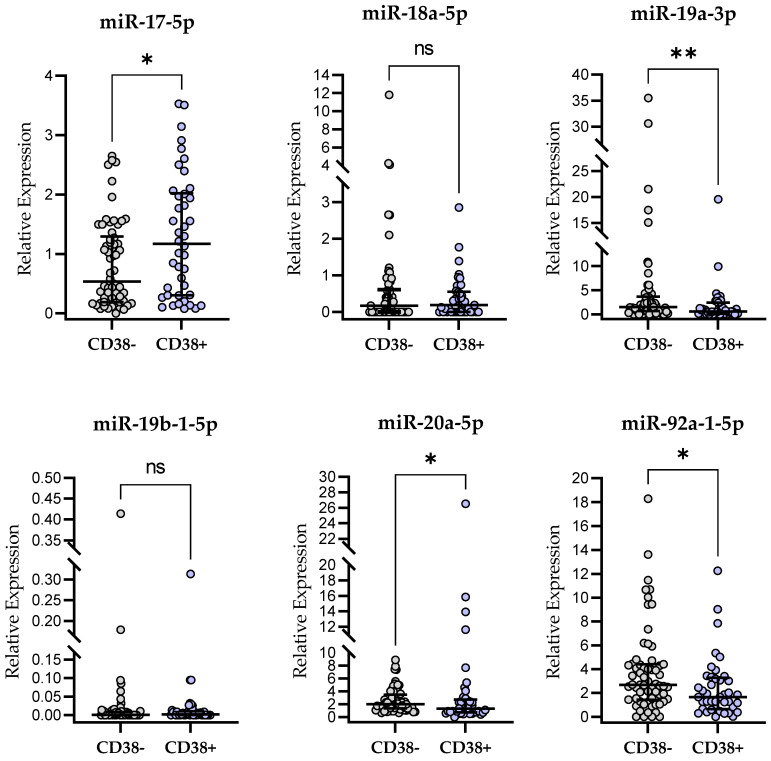
miR-17∼92-cluster expression in CD38-positive and CD38-negative patients. The central line shows the median and “whiskers” represent IQR. Scatter plots present raw data. The central line shows the median. “Whiskers” represent from the first quartile to the third quartile (IQR, interquartile range). * *p* < 0.05, ** *p* < 0.01; ns, not significant; the *p*-value was calculated using the U Mann–Whitney test. A cutoff point for CD38 positivity in CD19+/CD5+ cells was ≥30%.

**Figure 5 ijms-24-01705-f005:**
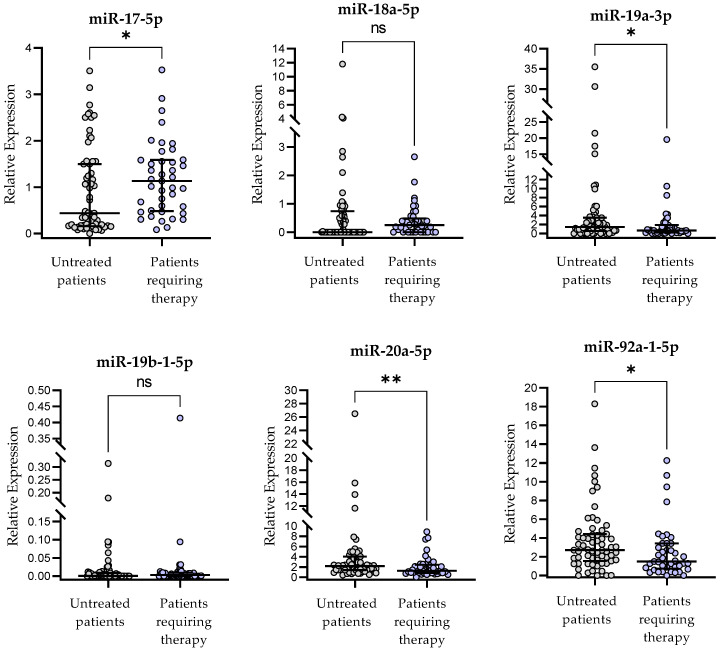
miR-17∼92-cluster expression in patients requiring therapy as compared to patients without treatment during the observation period. The central line shows the median and “whiskers” represent IQR. Scatter plots present raw data. The central line shows the median. “Whiskers” represent from the first quartile to the third quartile (IQR, interquartile range). * *p* < 0.05, ** *p* < 0.01; ns, not significant; the *p*-value was calculated using the U Mann–Whitney test.

**Figure 6 ijms-24-01705-f006:**
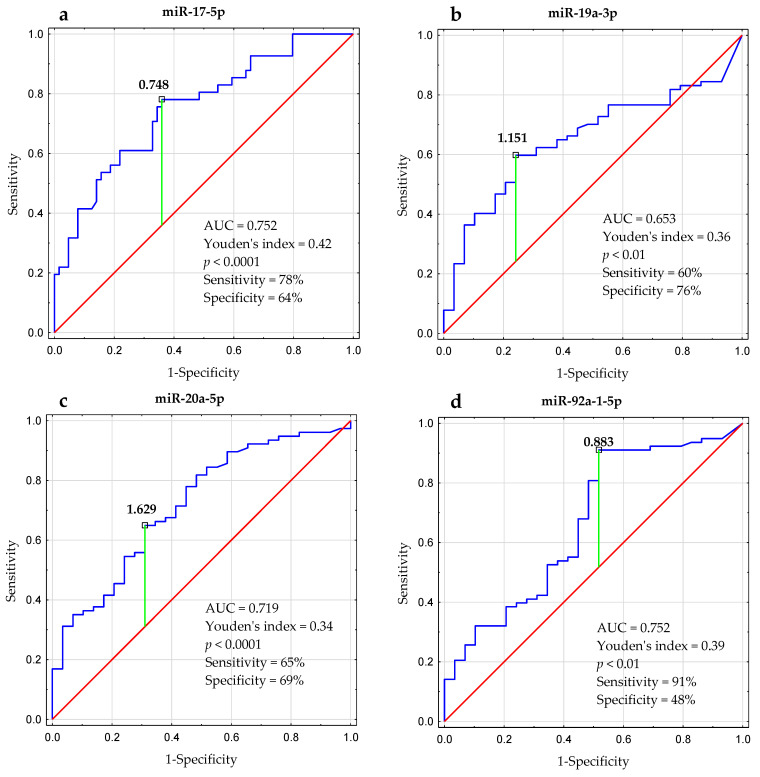
ROC (receiver operating characteristics) analysis and Youden index method were used to calculate the most significant cutoff value of (**a**) miR-17-5p, (**b**) miR-19a-3p, (**c**) miR-20a-5p, and (**d**) miR-92a-1-5p expression that best distinguished ZAP-70-positive and ZAP-70-negative cases. AUC (area under the curve) was also estimated.

**Figure 7 ijms-24-01705-f007:**
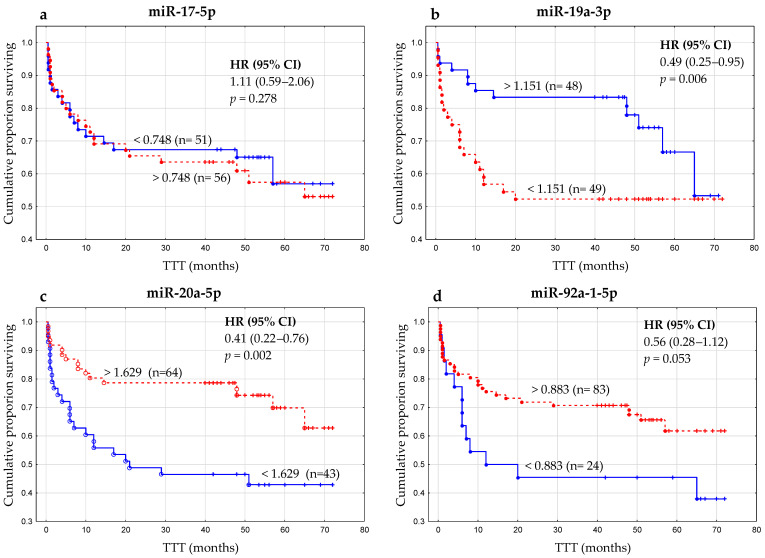
Kaplan–Meier curves based on the optimal thresholds for (**a**) miR-17-5 expression (>0.748), (**b**) miR-19a-3p (<1.151), (**c**) miR-20a-5p (<1.629), and (**d**) miR-92a-1-5p (<0.883) expression comparing TTT (time to treatment) among patients with CLL. HR, hazard ratio; CI, confidence interval.

**Table 1 ijms-24-01705-t001:** Baseline characteristics of CLL patients at the time of diagnosis.

Characteristics	n (%)
Total number of patients	107
Patient sex	
Female (%)	61 (57.0)
Male (%)	46 (43.0)
Rai Stage	
0 (%)	50 (46.7)
I (%)	18 (16.8)
II (%)	25 (23.4)
III (%)	5 (4.7)
IV (%)	9 (8.4)
Risk group	
Low	50 (46.7)
Intermediate	43 (40.2)
High	14 (13.1)
ZAP-70 (cutoff 20%)	
Positive (%)	30 (28.0)
Negative (%)	77 (72.0)
CD38 (cutoff 30%)	
Positive (%)	42 (39.3)
Negative (%)	65 (60.7)
Cytogenetic abnormalities	
del(17p13.1)	4 (3.7)
del(11q22.3)	16 (14.9)
Trisomy 12	9 (8.4)
sole del(13q14)	45 (42.0)
monoallelic del(13q)	35/45 (77.8)
biallelic del(13q)	10/45 (22.2)
Without del(17p13.1), del(11q22.3), trisomy 12, and del(13q14)	33 (31.0)
Patients requiring therapy (%)	45 (42.1)
Untreated patients (%)	62 (57.9)
	Median (IQR; interquartile range)
Age at diagnosis (years)	65 (61–72)
WBC count (G/L)	25 (17.15–54.79)
Lymphocyte count (G/L)	18.32 (10.56–44.96)
LDH (IU/L)	377 (325–424)
β2M (mg/dL)	2.46 (2.06–3.19)
CD19+/CD5+/ZAP-70+ cells (%)	7.85 (3.89–20.73)
CD19+/CD5+/CD38+ cells (%)	6.71 (16.05–37.04)

WBC, white blood cell; LDH, lactate dehydrogenase; and β2M, β2-microglobulin.

**Table 2 ijms-24-01705-t002:** miR-17∼92-cluster expression levels in CLL patients in different risk groups.

microRNAs	Low-Risk (Stage 0) n = 50	Intermediate-Risk (Stages I–II) n = 43	High-Risk (Stages III–IV) n = 14
miR-17-5p	0.466 (0.175–1.329) *	0.495 (0.247–1.511)	1.097 (0.293–1.945) *
miR-18a-5p	0.000 (0.000–0.555)	0.2955 (0.000–0.624)	0.274 (0.153–1.075)
miR-19a-3p	1.568 (0.000–0.009) *	0.8878 (0.214–2.944)	0.592 (0.000–1.374) *
miR-19b-1-5p	0.0007 (0.000–0.009)	0.0012 (0.000–0.006)	0.006 (0.000–0.015)
miR-20a-5p	2.031 (1.111–3.722) *	1.626 (0.832–3.204)	1.372 (1.070–2.348) *
miR-92a-1-5p	2.648 (1.401–4.739)	1.897 (0.645–3.552)	2.103 (0.887–4.378)

** p* < 0.05; The data are presented as the median and interquartile range (IQR: 25% percentile and 75% percentile); the *p*-value was calculated using the Kruskal–Wallis test with post-hoc Dunn’s correction.

**Table 3 ijms-24-01705-t003:** miR-17∼92-cluster expression levels in CLL patients responding to treatment and those with progressive disease.

Criterion	miR-17-5p	miR-18a-5p	miR-19a-3p	miR-19b-1-5p	miR-20a-5p	miR-92a-1-5p
Response (CR or PR)						
Median	0.680 ^##^	0.315	0.665	0.003	1.213 ^^	2.381 **
IQR	0.419–1.192	0.046–0.404	0.79–3.248	0.000–0.013	0.686–2.918	0.483–4.130
Stable Disease (SD)						
Median	0.754	0.268	0.885	0.0005	1.094	2.342
IQR	0.412–1.812	0.00–1.349	0.07–5.893	0.0001–0.003	0.766–2.988	0.598–8.560
Progressive Disease (PD)						
Median	1.360 ^#^	0.276	0.676	0.003	1.097 ^	1.289 *
IQR	0.595–1.589	0.000–0.719	0.511–1.835	0.0001–0.012	0.832–2.511	0.367–2.186
Deaths						
Median	1.676 ^#^	0.048	0.374	0.004	0.664 ^	0.658 **
IQR	0.869–2.208	0.000–0.572	0.000–1.261	0.000–0.014	0.000–1.093	0.087–1.824

IQR, interquartile range; CR, complete response; PR, partial response; ^#^
*p* < 0.05, ^# ^*p* < 0.05, ^ *p* < 0.05, ^ *p* < 0.05, * *p* < 0.05, *
*p* < 0.05; The *p*-value was calculated using the Kruskal–Wallis test with post-hoc Dunn’s correction.

**Table 4 ijms-24-01705-t004:** Univariate and multivariate analysis for time to treatment.

	Univariate			Multivariate	
Variable	Median TTT (Months)	HR (95% CI)	*p*	HR (95% CI)	*p*
Age					
≥65 years	46	2.01 (1.083–3.73)	**0.023**	1.71 (0.81–3.5)	0.159
<65 years	50				
ZAP-70					
≥20%	29	2.86 (1.52–5.38)	**0.008**	0.72 (0.31–1.65)	0.054
<20%	48				
CD38					
≥30%	44	1.67 (0.90–3.08)	**0.042**	0.95 (0.41–2.19)	0.434
<30%	48				
β2M					
≥3.5 mg/dL	16	2.65 (1.39–5.08)	**0.003**	2.30 (1.09–4.87)	**0.008**
<3.5 mg/dL	44				
del(17p13.1) or del(11q22.3)					
Positive	12	2.46 (1.29–4.73)	**0.006**	0.39 (1.09–4.87)	**0.039**
Negative	49				
del(13q14)					
Positive	49	0.51 (0.26–0.96)	**0.039**	0.68 (0.33–1.42)	0.31
Negative	42				
miR-17-5p					
≥0.748	42	1.11 (0.59–2.06)	0.278	NA	
<0.748	50				
miR-19a-3p					
≥1.151	49	0.49 (0.25–0.95)	**0.006**	1.67 (0.76–3.63)	0.195
<1.151	41				
miR-20a-5p					
≥1.629	50	0.41 (0.22–0.76)	**0.002**	2.96 (1.38–6.38)	**0.006**
<1.629	21				
miR-92a-1-5p					
≥0.883	48	0.56 (0.28–1.12)	0.053	NA	
<0.883	20				

TTT, time to treatment; β2M, β2 microglobulin; HR, hazard ratio; 95% CI: 95% confidence interval. NA, not assessed; The *p*-values < 0.05 are shown in bold. Only variables with *p* < 0.05 in the univariate analysis were added to the multivariate analysis.

## Data Availability

The data presented in this study are available within the article. Other data that support the findings of this study are available upon request from the corresponding authors.

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
