# Peer review of "Prognostic Value of the miR-17~92 Cluster in Chronic Lymphocytic Leukemia"

_ijms, 2023, doi:10.3390/ijms24021705_

Round 1

Reviewer 1 Report

The manuscript examines the effect of miR-17~92 cluster members on time to treatment (TTT) and overall survival (OS) in 107 newly diagnosed, untreated patients with CLL. All patients included in the study are precisely staged and classified. The basic advantage of the investigation is the adequate design and the variety of assays aimed to isolate and characterize PBMCs (Magnetic antibody cell separation of CD19+ B-cells, Flow cytometric analysis of CD38 and ZAP-70 expression, I-FISH) and to demonstrate the prognostic value of the miRNAs of interest.

The miR-17~92 cluster is examined in different risk groups, as well as in association with genetic aberrations, and ZAP-70 and CD 38 expression. The cluster is correlated with the clinical outcome of CLL patients and showed diverse pattern of expression. Six miRNAs were studied, three of which showed no significant differences between risk groups.

The original impact of the research is in the finding of association between monoallelic 13q deletion and the lower expression of miR-17-5p in leukemic lymphocytes compared to biallelic deletions. Patients with biallelic del (13q14) showed higher miR-20a-5p expression in leukemic B cells. Univariate and multivariate analyses revealed that low miR-20a expression is an independent marker predicting short TTT for CLL patients.

miR-17-5p expression increased with the disease stage, while high expression levels of miR-19a-3p and miR-20a-5p were detected in the low-risk group. The authors provide additional evidence to the assumption that miR-20a-5p is crucial in cancer development and outcome and it could serve as a  biomarker in CLL as well.

The expression heterogeneity is a fact to be considered when analyzing data as miR-19a-3p, miR-20a-5p, and miR-92a-1-5p showed higher levels in the ZAP-70 and CD38-negative group compared to ZAP-70 and CD38-positive patients.  Also, miR-17-5p was more expressed in ZAP-positive individuals compared with ZAP-70-negative ones.

The results of the study and presented in an informative and well documented way.  The discussion is through, focused and convincing. The conclusions correspond to the real results of the study. The entire manuscript is written in a good language style.

Based on the overall assessment of the study I could recommend it for publication in IJMS without further corrections.

Author Response

Dear Reviewer 1, thank you very much for reviewing our work. We so appreciate you for this high evaluation of this manuscript.

Reviewer 2 Report

The manuscript entitled "Prognostic value of the miR-17~92 cluster in chronic lymphocytic leukemia" is an original study on the prognostic role of miR-17∼92 cluster in CLL patients. The authors studied the expression of individual member in miR-17∼92 cluster including miR-17, miR-18a, miR-19a, miR-19b-1, miR-20a, and miR-92a-1 from CD19+ B lymphocytes.

The authors have found that higher expression of miR-17 was associated with unfavorable prognostic factors such as CD38 and ZAP-70 while higher expression of miR‐19a, miR‐20a, or miR‐92a was associated with better prognosis. Only miR-20a-5p was suggested as an independent factor for prognosis in the multivariate analysis.

Major issues:

The authors assessed microRNA expression in CD19+ B lymphocytes and evaluated the ZAP-70/CD38 expression in CD19+ and CD5+ lymphocytes. Although CLL usually expresses CD5, some CLL patients are still CD5 negative. Additionally, the expression between CD5 and ZAP-70/CD38 are not mutually exclusive. This undermines the correlation between miR-17∼92-cluster expression and ZAP-70/CD38 expression.

Minor issues:

1. The central line representing the median and the whiskers representing IQR are difficult to identify and read, especially in Figure 3, 4, and 5.

2. The comma is used as a decimal separator in Table 3 and Figure 6, which is inconsistent to the rest of the manuscript.

Author Response

Dear Reviewer 2, thank you very much for your valuable suggestions and comments. Indeed, in its past form, our manuscript suffered from several weaknesses. We improved our work according to your remarks. We believe that this paper will meet your expectations. Please find the more detailed answers to your suggestions below.

Major issues:

Point 1. The authors assessed microRNA expression in CD19+ B lymphocytes and evaluated the ZAP-70/CD38 expression in CD19+ and CD5+ lymphocytes. Although CLL usually expresses CD5, some CLL patients are still CD5 negative. Additionally, the expression between CD5 and ZAP-70/CD38 are not mutually exclusive. This undermines the correlation between miR-17∼92-cluster expression and ZAP-70/CD38 expression.

Response 1: In our study, patients were included in the analysis if their blood contained a CD5 positive B cell population (CD19+/CD5+) identified by flow cytometry. The sentence “All of CLL cases (100%) included in the analysis were CD5-positive and contained CD19+/CD5+ population (median (IQR) 79.05 (70.47-86.76)%)” was added (line: 79-81). In the "Materials and Methods" (line: 431-433) the sentences: “The purity of isolated CD19+ B cells was assessed by flow cytometry and was usually greater than 95%. Moreover, over 92% of selected B cells showed the CD19+/CD5+ phenotype" were added. In addition, in the revised version of our manuscript we added information about ZAP-70 and CD38 expression “In 81 patients (75.7%), there was a complete concordance of ZAP-70 and CD38 expression, i.e. 23 patients (21.5%) were positive for ZAP-70 and CD38, and 58 patients (54.2%) showed the ZAP-70-/CD38- phenotype. Additionally, 7 patients (6.5%) were characterized by the ZAP-70+/CD38- phenotype and 19 patients (17.8%) showed the ZAP-70-/CD38+ phenotype. Moreover, the percentage of CD19+/CD5+/CD38+ cells positively correlated with the percentage of leukemic cells expressing ZAP-70 (r = 0.486; p < 0.0001)” (line: 87-93).

Minor issues:

Point 1. The central line representing the median and the whiskers representing IQR are difficult to identify and read, especially in Figure 3, 4, and 5.

Response 1: The figures were changed. The lines representing the median with IQR went on top.

Point 2. The comma is used as a decimal separator in Table 3 and Figure 6, which is inconsistent to the rest of the manuscript.

Response 2: The comma has been changed to a dot.
